# Pemafibrate Prevents Retinal Pathological Neovascularization by Increasing FGF21 Level in a Murine Oxygen-Induced Retinopathy Model

**DOI:** 10.3390/ijms20235878

**Published:** 2019-11-23

**Authors:** Yohei Tomita, Nobuhiro Ozawa, Yukihiro Miwa, Ayako Ishida, Masayuki Ohta, Kazuo Tsubota, Toshihide Kurihara

**Affiliations:** 1Department of Ophthalmology, School of Medicine, Keio University, Shinjuku-ku, Tokyo 160-8582, Japan; y.tomita@keio.jp (Y.T.); n.ozw@z5.keio.jp (N.O.); yukihiro226@gmail.com (Y.M.); ayakoishida9696@gmail.com (A.I.); 2Laboratory of Photobiology, School of Medicine, Keio University, Shinjuku-ku, Tokyo 160-8582, Japan; 3Kowa Company, Ltd., Tokyo 160-8582, Japan; ms-ota@kowa.co.jp

**Keywords:** diabetes retinopathy, selective peroxisome proliferator-activated receptor alpha modulator (SPPARMα), fibroblast growth factor 21 (FGF21), hypoxia-inducible factor (HIF), vascular endothelial growth factor (VEGF)

## Abstract

Large-scale clinical trials, such as the Fenofibrate Intervention and Event Lowering in Diabetes (FIELD) and the Action to Control Cardiovascular Risk in Diabetes (ACCORD) studies, have shown that the administration of fenofibrate, a peroxisome proliferator-activated receptor alpha (PPARα) agonist, suppresses the progression of diabetic retinopathy. In this paper, we reveal a therapeutic effect of a selective PPARα modulator (SPPARMα), pemafibrate, against pathological angiogenesis in murine models of retinopathy. Oxygen-induced retinopathy (OIR) was induced in C57BL/6J mice by exposure to 85% oxygen from postnatal day eight (P8) for 72 h. Vehicle, pemafibrate or fenofibrate was administrated by oral gavage from P12 to P16 daily. Administration of pemafibrate, but not fenofibrate, significantly reduced pathological angiogenesis in OIR. After oral pemafibrate administration, expression levels of downstream PPARα targets such as acyl-CoA oxidase 1 (*Acox1)*, fatty acid binding protein 4 (*Fabp4),* and fibroblast growth factor 21 (*Fgf21)* were significantly increased in the liver but not in the retina. A significant increase in plasma FGF21 and reduced retinal hypoxia-inducible factor-1α (HIF-1α) and vascular endothelial growth factor A (*Vegfa*) were also observed after this treatment. In an in vitro HIF-luciferase assay, a long-acting FGF21 analogue, but not pemafibrate, suppressed HIF activity. These data indicate that SPPARMα pemafibrate administration may prevent retinal pathological neovascularization by upregulating FGF21 in the liver.

## 1. Introduction

Diabetic retinopathy (DR) and age-related macular degeneration (AMD) are major causes of vision impairment worldwide. The number of diabetic patients increases every year, so the prevention of DR onset has become an increasingly important issue in many countries [1]. These diseases are characterized by pathological angiogenesis [1,2]. In the last few decades, anti-vascular endothelial growth factor (VEGF) therapy has been established to preserve and maintain the visual function in these neovascular retinal diseases [3,4,5]. As a pathophysiological mechanism of DR, retinal ischemia has been shown to produce angiogenic and inflammatory agents including VEGF, which induce the activity of inflammatory leukocytes, and later cause pathological angiogenesis [6]. Anti-VEGF therapy is now being applied to diabetic macular edema, as well as proliferative retinopathy. However, anti-VEGF therapy still faces several difficulties, such as affordability, difficulty with visual acuity improvement, and potential for geographic atrophy after prolonged exposure to treatment [7]. Therefore, developing an alternative therapeutic treatment is necessary.

Besides the pathological role of retinal ischemia in DR, metabolic changes in the retinal tissues due to hypoxic conditions were also reported and may contribute to the onset of AMD [8]. Lipid deposition has also been shown to play a role in AMD pathogenesis. Throughout aging, lipid metabolites deposited and peroxidized under retinal pigment epithelial (RPE) cells may cause chronic inflammation and lead to the onset of AMD [9,10]. Therefore, the regulation of lipid metabolism is important to consider when addressing this disease.

According to two large randomized controlled trials, the Fenofibrate Intervention and Event Lowering in Diabetes (FIELD) study [11] and the Action to Control Cardiovascular Risk in Diabetes (ACCORD) study [12], a fibric acid derivative, fenofibrate, inhibits the progression of DR and other microvascular endpoints in patients with type 2 diabetes. Fenofibrate is primarily used as a hypolipidemic agent, which largely reduces triglyceride levels and increases high-density lipoprotein (HDL)-cholesterol levels, by activating peroxisome proliferator-activated receptor alpha (PPARα). Although activation of PPARα by fenofibrate has been shown to suppress pathological angiogenesis, precise mechanisms of a PPARα activator fenofibrate in DR are not yet fully understood [13].

In recent years, a selective PPARα modulator (SPPARMα) pemafibrate (Parmodia^®^, K-877, Kowa, Tokyo, Japan) was approved in Japan in July 2017 as a therapeutic drug against hyperlipidemia. Pemafibrate reduces serum triglycerides (TGs) and increases the HDL cholesterol in a similarly way to fenofibrate [14]. Unlike the renal excretion of fenofibrate, pemafibrate is metabolized in the liver, and consequently can be used in patients with mild renal impairment. In addition, pemafibrate has been shown to activate PPARα more specifically than fenofibrate [15]. However, the therapeutic effects of pemafibrate in ocular diseases like AMD and DR have not yet been reported.

Pemafibrate has been reported to activate PPARα in hepatocytes resulting in an increase of plasma fibroblast growth factor 21 (FGF21) levels [16]. FGF21 is a secreted protein composed of 209 amino acids that was first reported in 2000 [17]. Systemic administration of FGF21 has favorable effects on glucose and lipid metabolism in mice and improves the level of blood lipids in type 2 diabetic patients [18]. Recently, a long-acting FGF21 was shown to suppress pathological neovascularization in the retina and choroid [19], thereby exerting a protective effect on the neural retina [20].

In this study, using the mouse oxygen-induced retinopathy (OIR) model, which is the best characterized and widely used in vivo retinal neovascularization model [21,22], we examined whether the administration of pemafibrate has an inhibitory effect on pathological angiogenesis. We explored the mechanism underlying the induction of FGF21 expression. Through the results of this study we wanted to uncover future potential therapeutic reagents to address prevalent retinopathies.

## 2. Results

### 2.1. Antiangiogenic Effect of Oral Administration of Pemafibrate on the Retina of OIR

To investigate the effect of pemafibrate on angiogenesis, we used the retinas of OIR model mice (Figure 1A–F). We found no significant differences in body weight among the groups at postnatal day 12 (P12) or P17 (Figure 1G). There were no significant differences in vaso-obliteration (VO) among the groups (*p* = 0.19) (Figure 1H). The neovascular tufts (NV) area in the pemafibrate group was significantly decreased compared with the vehicle group; however, no significant changes were found between the fenofibrate and the vehicle groups (Figure 1I). These data indicate that oral administration of pemafibrate prevents pathological but not physiological retinal neovascularization.

### 2.2. Pemafibrate Directly Acts in the Liver and Promotes Expression of Factors Downstream of PPARα

Next, we explored the primary target organ of the drug. In the retina, no significant differences occurred in expression between the pemafibrate and the vehicle groups for genes downstream of PPARα, including acyl-CoA oxidase 1 (*Acox1*), fatty acid binding protein 4 (*Fabp4*), and *Fgf21* (Figure 2A–C). In contrast, the mRNA expression levels of these genes were significantly higher in the liver of the pemafibrate group compared with the vehicle group (Figure 2D–F). These data suggest that oral administration of pemafibrate directly affects the liver but not the retina.

### 2.3. Pemafibrate Increases Plasma FGF 21 Concentration and Suppresses Expression of Vegfa in the Retina

We focused on FGF21 as its mRNA expression was increased in the liver after pemafibrate administration. The plasma FGF21 concentration was significantly elevated in the pemafibrate and fenofibrate group (P13) compared with the control group (Figure 3A). The mRNA expression level of *Fgf21* was significantly increased in the pemafibrate not fenofibrate (Figure 3B). The mRNA expression level of *Vegfa* significantly decreased in the pemafibrate and fenofibrate group compared with the vehicle group (Figure 3C). These data suggest that elevated plasma FGF21 may be involved in the inhibition of *Vegfa* within the retina.

### 2.4. Oral Administration of Pemafibrate Inhibits the Retinal Expression of HIF-1α

To evaluate the mechanism underlying the suppression of *Vegfa* expression, we focused on hypoxia-inducible factor-1α (HIF-1α), which is a transcription factor controlling *Vegfa* gene expression. Using immunohistochemistry, we detected a decrease in the expression of HIF-1α in the retinal inner layer in the OIR model mouse pemafibrate group (Figure 4). These data demonstrate that oral administration of pemafibrate suppresses retinal HIF-1α expression in OIR model mice.

### 2.5. Inhibitory Effect of FGF21 on HIF Activity in Vitro

To examine HIF inhibitory effects of pemafibrate and a long-acting FGF21 molecule (PF-05231023), a HIF luciferase assay was performed in vitro. Pemafibrate or PF-05231023 was added to the 661W cell line, which has properties of both retinal ganglion and photoreceptor cells [23], whereas pseudohypoxia was induced with CoCl_2_. PF-05231023 (0.2 and 2 nM) significantly inhibited HIF activity compared with CoCl_2_ alone (Figure 5A). In contrast, administration of pemafibrate with CoCl_2_ showed no inhibitory effect on HIF activity (Figure 5B). These data suggest that systemic administration of pemafibrate might indirectly inhibit retinal HIF activity by elevating the plasma FGF21 concentration.

## 3. Discussion

In the current study, we revealed that oral administration of a selective PPARα modulator, pemafibrate, inhibited pathological angiogenesis in the retina of OIR model mice (Figure 1). A significant increase in PPARα target gene expression in the liver (Figure 2) and an elevation of the plasma FGF21 concentration (Figure 3) was observed after systemic administration of pemafibrate, which is consistent with a previous report of pemafibrate increasing FGF21 expression in patients with type 2 diabetes with hypertriglyceridemia and in high fat diet mice [24,25]. The authors showed that the elevation of FGF21 after pemafibrate administration was not observed in PPARα knockout mice [25].

Recently a long-acting FGF21 molecule, PF-05231023, was reported to have anti-angiogenic activity in the retina of OIR model mice [19]. In this report, they showed that increased adiponectin expression after PF-05231023 administration produced anti-angiogenic effects through the inhibition of tumor necrosis factor (TNF)-α. However, in our experiment, we found no difference in the mRNA expression level of adiponectin and TNF-α between the vehicle and pemafibrate groups, in the OIR model (data not shown).

A study reported that fenofibrate prevents pathological neovascularization in the rat OIR model by suppressing HIF and VEGF [26]. HIF plays an important role in maintaining tissue homeostasis after exposure to hypoxic conditions and other stress stimuli [27,28]. Thus, we also focused on the HIF/VEGF system as a potential pathway activated by pemafibrate. In addition, HIF stabilization induces target gene expression, such as *Vegfa* and pyruvate dehydrogenase lipoamide kinase isozyme 1 (*Pdk1*), for anaerobic metabolism [29,30]. In fact, we have previously found that multiple HIF-responsive genes including *erythropoietin*, *angiopoietin-2*, and *Pdk1,* in addition to *Vegfa,* were upregulated in the retina of OIR model mice [31]. In this current experiment, administration of pemafibrate suppressed the mRNA expression of *Vegfa* in the retina of P13 OIR model mice (Figure 3). Pemafibrate suppressed HIF-1α expression in the inner retina of OIR model mice (Figure 4). From the results of the HIF luciferase assay, PF-05231023 suppressed HIF activity in retinal photoreceptor cells. In contrast, pemafibrate did not show a HIF inhibitory effect. Taken together, the direct target organ of systemic administration of pemafibrate is presumably the liver, which can produce FGF21 to suppress HIF activity, and thereby inhibit retinal pathological angiogenesis (Figure 6).

In the current study, pemafibrate, but not fenofibrate, protected against pathological retinal neovascularization (Figure 1). Studies have shown that fenofibrate administration with a similar dose to the current study had an inhibitory effect against neovascularization in OIR model mice [32,33]. The disagreement in results may be due to differences between intervention routes or experimental protocols. Both chemicals showed similar trends of increased levels of plasma FGF21 and upregulation of *Fgf21* expression in the liver, although the result of fenofibrate administration had high variation (Figure 3). Although both fenofibrate and pemafibrate affect PPARα, the effects are different. First, fenofibrate induces not only PPARα but also PPARδ, whereas pemafibrate induces only PPARα, by binding to the entire cavity region of ligand-binding pocket [34]. Second, PPARα agonists have the potential to trigger different biological responses via the same receptor [35]. As such, these findings may indicate that pemafibrate is more powerful than fenofibrate for the prevention of neovascularization. A meta-analysis of RCTs showed that in humans, pemafibrate exhibits significantly fewer adverse events and, an increase in creatinine levels was significantly lower than fenofibrate [36]. These may be the reasons why pemafibrate and fenofibrate have different effects for the retina.

## 4. Materials and Methods

### 4.1. Ethics Statement

All animal studies adhered to the Association for Research in Vision and Ophthalmology Statement for the Use of Animals in Ophthalmic and Vision Research, from the National Institutes of Health (NIH) guidelines for work with laboratory animals, “Animal Research: Reporting of in vivo Experiments (ARRIVE)” guidelines, and were approved (#16017-1) on 25 October 2017 by the Ethics Committee for Animal Research of Keio University School of Medicine (Tokyo, Japan).

### 4.2. Mice

C57BL/6J mice were obtained from CLEA Japan, Inc. OIR was induced in the mice by exposure to 85% oxygen from postnatal day 8 (P8) for 72 h. Pemafibrate (0.3 mg/kg/day; Kowa, Tokyo, Japan), fenofibrate (10 mg/kg/day; Sigma-Aldrich, St. Louis, MO, USA) and vehicle (methyl cellulose: Shin-Etsu Chemical, Tokyo, Japan) were fed via oral gavage from P12 to P16 daily. Mice under normoxia with vehicle were also prepared (NOX group). At P17, the vaso-obliteration (VO) and neovascular tuft (NV) areas were evaluated on the wholemount retinae with iso-lectin B4 staining, as previously described [21].

### 4.3. Real-Time PCR

The expression levels of various genes were measured in the retina and the liver from the sacrificed animals at P13 and P17.

The total mRNA of the retina was extracted from the mouse retinal samples using RNeasy Plus Mini kit (Qiagen, Velno, Netherlands), and reverse-transcribed using ReverTra Ace qPCR RT Master Mix (TOYOBO, Osaka, Japan). Real-time RT-PCR was performed using THUNDERBIRD SYBR qPCR Mix (TOYOBO, Osaka, Japan). Data were analyzed with StepOne, Software version 2.3 (Applied Biosystems, Waltham, MA, USA).

The total mRNA of the liver was extracted from the mouse liver homogenate following the manufacturer’s instructions using ISOGEN II (Nippon Gene, Tokyo, Japan). Reverse transcription of the extracted mRNA was performed using High-capacity cDNA Reverse Transcription Kits (Life Technologies, Carlsbad, CA, USA) in accordance with their instructions. Fast SYBR^®^ Green Master Mix (Applied biosystems, Waltham, MA, USA) was used as the reagent, and the Fast Real-Time PCR System (7900HT, Applied Biosystems, Waltham, MA, USA) was used for quantitative PCR. The primer sequences for gene expression analysis were: mouse *β-actin*: forward, 5′-GGAGGAAGAGGATGCGGCA-3′, reverse, 5′-GAAGCTGTGCTATGTTGCTCTA-3′, mouse *Fgf21*: forward, 5′-ACAGCCATTCACTTTGCCTGAGC-3′, reverse, 5′-GGCAGCTGGAATTGTGTTCTGACT-3′, mouse *Acox1*: forward, 5′-TCTTCTTGAGACAGGGCCCAG-3′, reverse, 5′-GTTCCGACTAGCCAGGCATG-3′, mouse *Fabp4*: forward, 5′-CCGCAGACGACAGGA-3′, reverse, 5′-CTCATGCCCTTTCATAAACT-3′, mouse *Vegfa*: forward, 5′-CCCTCTTAAATCGTGCCAAC-3′, reverse, 5′-CCTGTCCCTCTCTCTGTTCG-3′. β-actin was used as an internal control.

### 4.4. Measurement of Plasma FGF21

Concentrations of FGF21 in the plasma was measured at P13 and P17. The plasma FGF21 levels were measured with a microplate reader (VersaMax, Molecular Devices, San Jose, CA, USA) using the FGF-21 ELISA kit (#RD291108200R, BioVendor Laboratory Medicine, Brno, Czech Republic) according to the manufacturer’s protocol.

### 4.5. Immunohistochemistry

Immunohistochemistry (IHC) for the retina was performed as previously described [37]. The primary antibody for HIF-1α was established by immunizing purified fusion proteins encompassing amino acids 416 to 785 of mouse HIF-1α, into guinea pigs [37]. Alexa Fluor 488 conjugated IgG (Life Technologies) was used as the secondary antibody. Nuclei were stained with DAPI (Fluoromount-G, SouthernBiotech, Birmingham, AL, USA). The images were captured with a fluorescence microscope (BZ 9000, Keyence, Osaka, Japan).

### 4.6. HIF-Luciferase Assay

To monitor HIF transcriptional activity, the mouse 661W cell line, which has aspects of both photoreceptor and retinal ganglion cells, was transfected with a HIF-luciferase reporter gene construct (Cignal Lenti HIF reporter, Qiagen). The luciferase construct encodes the firefly luciferase gene under the control of a hypoxia-related element that connects to HIF. CMV-Renilla luciferase was also co-transfected into these cells as an internal control. q before luminescence was measured, HIF activation of normal oxygen pressure was induced by administering cobalt chloride (CoCl_2_, 200 μM, cobalt (II) chloride hexahydrate, FUJIFILM Wako Pure Chemical Corporation, Osaka, Japan) to the cells. To evaluate the inhibitory effect of a long-acting FGF21 molecule (PF-05231023, MedKoo Biosciences, Morrisville, NC, USA) and pemafibrate on HIF activation, these two drugs were added at the same time as CoCl_2_ administration. Luminescence was measured with Dual Luciferase Reporter Assay System (Promega, Madison, WI, USA). The suppressive effect of pemafibrate and PF-05231023 against HIF activation was evaluated 24 h after the induction.

### 4.7. Statistical Analysis

All data are expressed as means ± standard error. Statistical significance was calculated using Student’s *t*-test or 1-way ANOVA and Tukey’s multiple comparison tests with Prism 6 (GraphPad Software, San Diego, CA, USA).

## 5. Conclusions

SPPARMα pemafibrate alleviated pathological neovascularization in the retina by attenuating the HIF/VEGF pathway through the induction of *Fgf21* in the liver, but not in the retina. Although further investigations are required to better understand the mechanism underlying this phenomenon, the current results indicate that oral administration of pemafibrate may be a promising therapeutic strategy to treat neovascular retinal diseases.

## Figures and Tables

**Figure 1 ijms-20-05878-f001:**
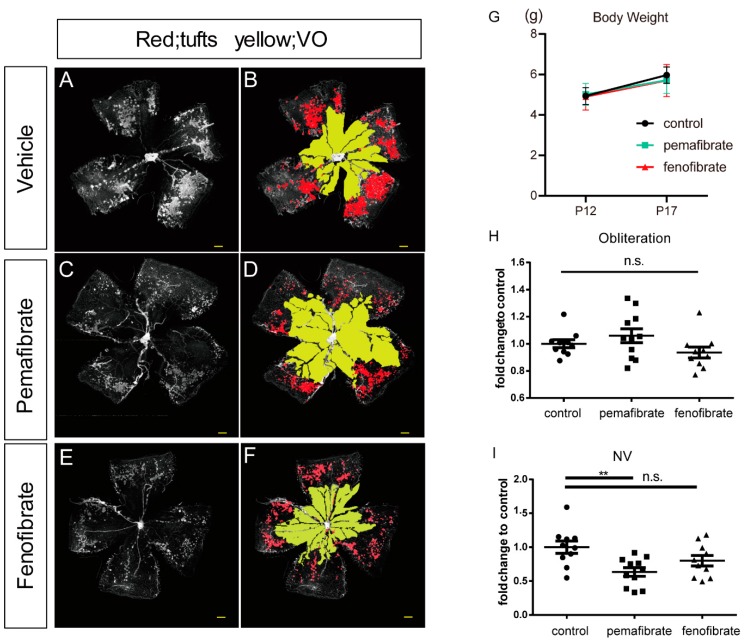
Pemafibrate has an anti-angiogenic effect in the retina. (**A**–**F**) Representative retinal images of the each oxygen-induced retinopathy (OIR) model mice (red, neovascular tufts (NV); yellow, vaso-obliteration (VO)), scale bar: 500 μm. (**G**) The change in the body weight among the groups (day 12 (P12) and P17, *n* = 6). (**H**) Quantification of VO area with each group (P17, *n* = 10,11). (**I**) Quantification of NV area with each group (P17, *n* = 10,11). Note that oral administration of pemafibrate prevents pathological but not retinal neovascularization. The data were analyzed by 1-way ANOVA and Tukey’s multiple comparison test and are expressed as mean ± standard error (SE). ** *p* < 0.01. n.s., not significant.

**Figure 2 ijms-20-05878-f002:**
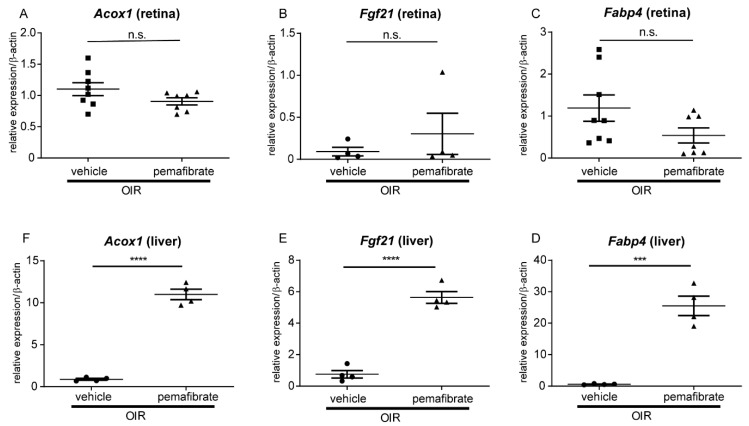
Pemafibrate stimulates peroxisome proliferator-activated receptor alpha (PPARα) downstream gene expression in the liver but not in the retina. (**A**–**C**) The mRNA expression levels of PPARα downstream genes including acyl-CoA oxidase 1 (*Acox1)*, fatty acid binding protein 4 (*Fabp4),* and fibroblast growth factor 21 (*Fgf21)* in the retina (P17, *n* = 7,8) and (**D**–**F**) in the liver (**D**–**F**; P17, *n* = 4) in OIR model mice. Note that oral administration of pemafibrate increased the targeted genes in the liver but not in the retina. The data were analyzed using Student’s *t*-test and are expressed as mean ± SE. *** *p* < 0.001; **** *p* < 0.0001. n.s., not significant.

**Figure 3 ijms-20-05878-f003:**
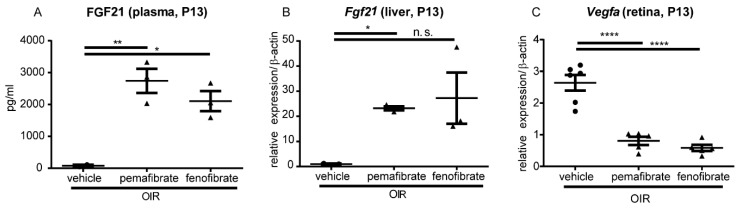
Pemafibrate and fenofibrate increases the FGF21 concentration in the plasma and suppress Vascular endothelial growth factor A (*Vegfa*) expression in the retina. (**A**) The plasma concentration of FGF21 in OIR model mice at P13 (*n* = 2,3). (**B**) The mRNA expression of *Fgf21* in the liver in OIR model mice at P13 (*n* = 3). (**C**) The mRNA expression of *Vegfa* in the retina in OIR model mice at P13 (*n* = 5,6). Note that oral administration of pemafibrate and fenofibrate increased the plasma level of FGF21 and suppressed the retinal expression of *Vegfa*. A significant upregulation of *Fgf21* mRNA in the liver was observed in the pemafibrate group, but not in fenofibrate group. The data were analyzed by 1-way ANOVA and Tukey’s multiple comparison tests and are expressed as mean ± SE. * *p* < 0.05, ** *p* < 0.01, **** *p* < 0.0001. n.s., not significant.

**Figure 4 ijms-20-05878-f004:**
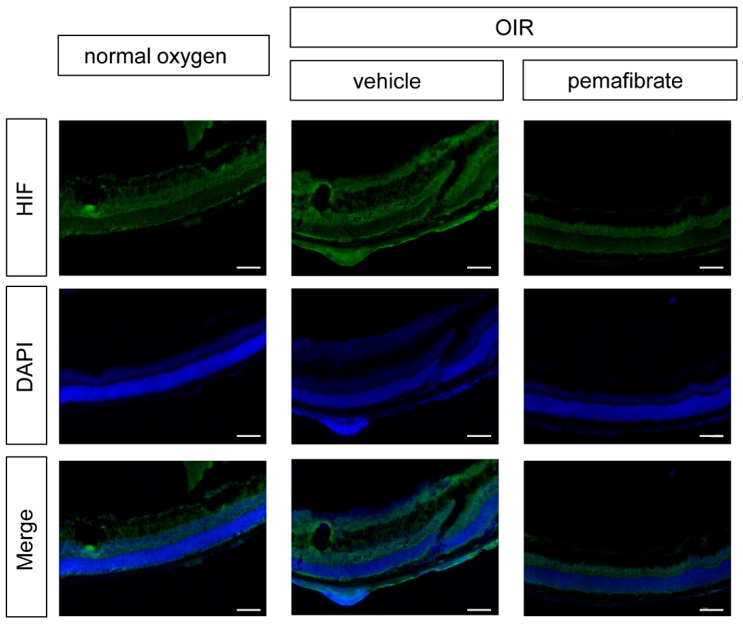
The systemic administration of pemafibrate inhibits hypoxia-inducible factor-1α (HIF-1α) in the retina. Ocular cross-sections of OIR model mice (P17) injected with pemafibrate and vehicle, and normoxia mice were immune-stained with antibodies specific for HIF-1α (green). Nuclei were counterstained with DAPI (blue). scale bar: 500 μm.

**Figure 5 ijms-20-05878-f005:**
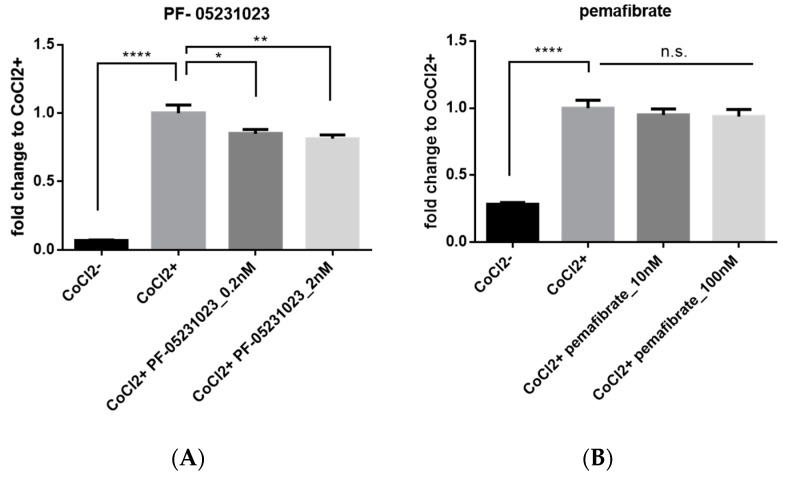
A long-acting FGF21 molecule but not pemafibrate has an inhibitory effect on HIF activity in vitro. HIF-reporter luciferase assay was performed with 661W cells. (**A**) A long-acting FGF21 molecule (PF-05231023; 0.2 and 2 nM) and (**B**) pemafibrate (10 and 100 nM) were added together with CoCl_2_. Note that PF-05231023 but not pemafibrate significantly inhibited HIF activity induced by CoCl_2_. The data were analyzed by one-way ANOVA and Tukey’s multiple comparison tests and are expressed as mean ± SE. * *p* < 0.05, ** *p* < 0.01, **** *p* < 0.0001). n.s., not significant.

**Figure 6 ijms-20-05878-f006:**
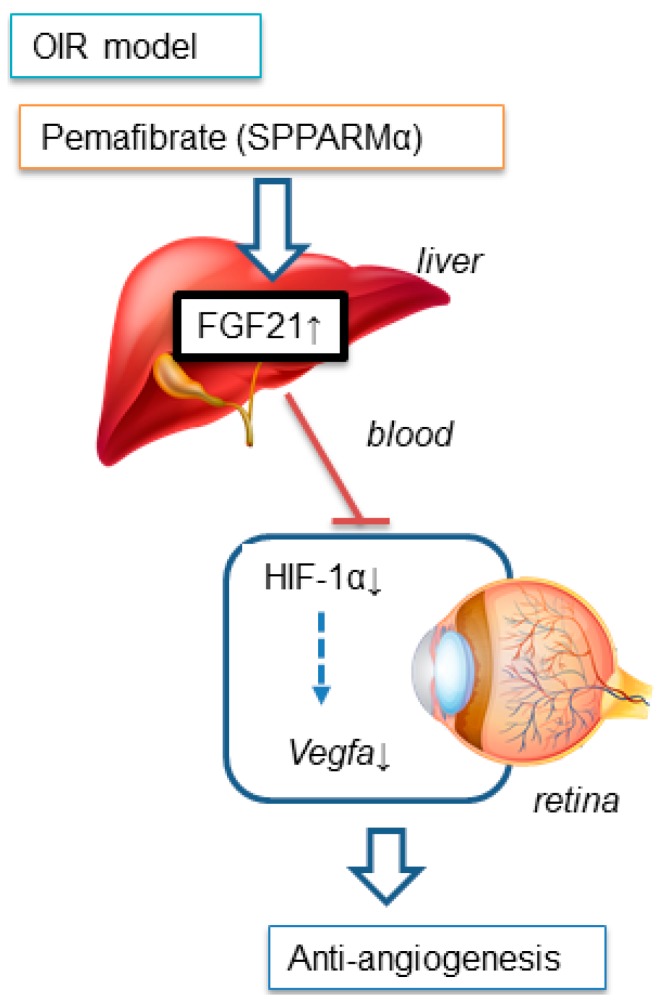
The suppressive mechanism of pathological retinal angiogenesis by systemic administration of pemafibrate. Systemic selective PPARα modulator (SPPARMα) pemafibrate administration upregulates PPARα target genes including *Fgf21* in the liver, but not in the retina. The increased level of the plasma FGF21 concentration resulting in the downregulation of *Vegfa* via HIF inactivation, may be a possible mechanism for prevention of retinal pathological neovascularization.

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
