# Peer review of "Pemafibrate Prevents Retinal Pathological Neovascularization by Increasing FGF21 Level in a Murine Oxygen-Induced Retinopathy Model"

_ijms, 2019, doi:10.3390/ijms20235878_

Round 1
Reviewer 1 Report
In this manuscript, the authors investigated the effect of pemafibrate, a PPARa activator, on retinal neovasculation in a mouse oxigen-induced retinopathy model. They also present the mechanism, by which pemafibrate modifies the naovasculation and angiogenesis. They show pemafibrate induces FGF21 in the liver and increases the level in blood, leading to the inhibition of HIF1 and VEGFa and to the reduction of neovascularization in the retina.
The manuscript can be improved if it is revised as I suggest below.
Major points
The connection of FGF21 and the effect of pemafibrate in the retina is not convincing to me from the presented data. There may be other proteins also produced in the liver of pemafibrate-treated mice to contribute to the effect. Are the effects of pemafibrate in neovascularization blocked by injections of an FGF21 inhibitor? Or can they use FGF21 knockout mouse to test the effect of pemafibrate?
At the end of discussion, the authors mention about the difference between pemafibrate and fenofibrate. However, through the entire manuscript, the effects of pemafibrate and fenofibrate were compared only in the tuft formation shown in Figure 1. Although the effect of pemafibrate was higher than fenofibrate, both showed similar tuft reduction. They should compare the two chemicals for FGF21 levels in blood and FGF21 mRNA expression in liver if the effect is specific for pemafibrate or not.
Minor points
In the Figure 1, the area of tuft in pemafibrate-treated mouse retina is not convincingly reduced. From the graph in Figure1, st least about 50% of the tuft formation should be reduced. The authors should show better results to convince readers.
Title of Figure 2 "Pemafibrate directly affects the liver but not the retina." The result does not show if the increase of gene expression in liver was directly or indirectly caused by pemafibrate. It may be changed to "Pemafibrate stimulates PPARa downstream gene expression in the liver but not in the retina."
Reviewer 2 Report
Tomita and coauthors evaluated the effects of pemafibrate, a PPARa activator, in inhibiting ocular neovascularization in a mouse model of oxygen-induced retinopathy (OIR). Previous clinical studies have found that PPARa activators, as lipid lowering drugs, are also effective in inhibiting diabetic retinopathy progression, yet the mechanisms are unclear. Here the authors found that oral gavage of pemafibrate increases liver (hence circulating) levels of FGF21, which then suppresses retinal HIF-1 and Vegfa to result in suppression of retinal neovascularization. Overall the research design is sound and data as presented are largely supportive of the conclusions. Yet Fig4 is missing, which must be provided in the revision to allow re-evaluation of the data quality. In addition, there are a few other concerns which may help strengthen the manuscript.
Major concerns:
Fig 4 is missing and must be provided in revision. If FGF21 regulates retinal HIF-1 levels, it will be helpful to evaluate other HIF target genes, including erythropoietin, angiopoietin-2, and others, in addition to VEGF. Showing that multiple HIF-responsive genes are upregulated in the retina will be more convincing. Line 85: “…no significant change between the fenofibrate and the vehicle group (Fig. 1I)”. Figure 1I only shows n.s. between pemafibrate and fenofibrate groups. What is the p-value between the fenofibrate and the vehicle group? If fenofibrate is not effective against vehicle, does that suggest other potentially PPARa-independent function of pemafibrate, if fenofibrate does not work? Methods: what are the doses of pemafibrate and fenofibrate used? Is the fenofibrate dose/intervention route different from previous studies suggesting protective effects of fenofibrate in OIR?Minor concerns:
Line 130: the authors noted decreased HIF1a expression in the inner retina. Then what is the rationale for using 661W cells, a photoreceptor cell line representing outer (but not inner) retina? Line 83: spell out VO and NV. Fig. 2, some panels are misaligned. Line 117: “we concluded that elevated plasma FGF21 is involved in the inhibition of Vegfa in the retina”. This statement is a bit premature and overstating, the evidence presented up to Fig. 3 in the manuscript does not yet support a causal relationship between plasma FGF21 and retinal Vegfa. This sentence can be toned down as “we then asked that where elevated plasma FGF21 may be involved in the inhibition of Vegfa in the retina”.Author Response
Please see the attachment.

Round 2
Reviewer 2 Report
All concerns are addressed.